# Expanding the Molecular Disturbances of Lipoproteins in Cardiometabolic Diseases: Lessons from Lipidomics

**DOI:** 10.3390/diagnostics13040721

**Published:** 2023-02-15

**Authors:** Christina E. Kostara

**Affiliations:** Laboratory of Clinical Chemistry, School of Health Sciences, Faculty of Medicine, University of Ioannina, 45110 Ioannina, Greece; chkostara@gmail.com

**Keywords:** lipidomics, lipoproteins, metabolism, cardiometabolic diseases, obesity, metabolic syndrome, prediabetes, diabetes, non-alcoholic fatty liver disease, cardiovascular disease, chronic renal failure, NMR

## Abstract

The increasing global burden of cardiometabolic diseases highlights the urgent clinical need for better personalized prediction and intervention strategies. Early diagnosis and prevention could greatly reduce the enormous socio-economic burden posed by these states. Plasma lipids including total cholesterol, triglycerides, HDL-C, and LDL-C have been at the center stage of the prediction and prevention strategies for cardiovascular disease; however, the bulk of cardiovascular disease events cannot be explained sufficiently by these lipid parameters. The shift from traditional serum lipid measurements that are poorly descriptive of the total serum lipidomic profile to comprehensive lipid profiling is an urgent need, since a wealth of metabolic information is currently underutilized in the clinical setting. The tremendous advances in the field of lipidomics in the last two decades has facilitated the research efforts to unravel the lipid dysregulation in cardiometabolic diseases, enabling the understanding of the underlying pathophysiological mechanisms and identification of predictive biomarkers beyond traditional lipids. This review presents an overview of the application of lipidomics in the study of serum lipoproteins in cardiometabolic diseases. Integrating the emerging multiomics with lipidomics holds great potential in moving toward this goal.

## 1. Introduction

Human plasma is estimated to consist of thousands of functionally and chemically diverse molecular lipid species [1], which are transported through the vascular system forming protein- and lipid-based spherical macromolecular complexes called lipoproteins. The most abundant amphipathic lipids (phospholipids and free cholesterol) form a surface monolayer, and the more hydrophobic or neutral lipids (cholesteryl esters and triglycerides) are placed in the particle core. Knowledge of the molecular organization of the lipid and protein constituents is essential to understand the alterations occurring in the structure of lipoproteins induced by the altered activity of enzymes involved in lipid metabolism.

A cluster of metabolic diseases is triggered by abnormalities in lipid metabolism such as dyslipidemia and obesity, culminating in the metabolic syndrome (MetS) and, furthermore, to prevalent chronic diseases including insulin-resistance states, type 2 diabetes (T2D), atherosclerosis, and cardiovascular disease (CVD). Both genetic and environmental factors (diet, nutrition, and lifestyle) are the cause of these lipid-related cardiometabolic diseases and they may exist long before becoming clinically apparent. Dysregulated lipid metabolism is recognized as an established risk factor in CVD. However, it is still largely unknown how and which lipid molecular species affect the risk of CVD and potentially improve CVD prediction in addition to the traditional serum lipid biomarkers [2]. Thus, there is an urgent clinical need for the identification of specific lipid signatures at early stages of disease to delay or prevent morbidity and mortality, if possible, via personalized treatment strategies based on a patient’s unique phenotype.

Comprehensive lipid profiling, or lipidomics, defines the chemical lipid phenotype of biological matrices; as such, it has a unique potential to unravel and elucidate the underlying aberrant molecular lipid-related pathways associated with the incidence, severity, and progression of cardiometabolic diseases [3,4]. Accordingly, lipidomics could play an essential role in defining the biochemical mechanisms underlying the lipid-related disease processes through the identification of the alterations in cellular lipid signaling and metabolism.

In this review, the first section focuses on the existing conventional serum lipid biomarkers used for the diagnosis of lipid disorders, monitoring the response of patients to lipid-lowering therapy and also for the assessment of CVD risk and the need for the shift to the comprehensive lipid profiling. This is followed by the detailed description of the steps comprising the lipidomics workflow, and finally, the last part of the review emphasizes the disturbances occurring in lipid composition of serum lipoproteins in cardiometabolic diseases.

## 2. From Conventional Lipid Biomarkers to Comprehensive Lipid Profiling

Disturbances in lipid metabolism play a central role in the onset and progression of cardiometabolic diseases [5,6,7]. In clinical practice, the mainstay of measurements for the assessment of dysregulations in lipid metabolism consists of two lipid classes in the bloodstream: triglycerides (triacylglycerols) and cholesterol carried by circulating lipoproteins. Specifically, the levels of triglycerides, total cholesterol, and cholesterol carried by high-density lipoprotein (HDL) (HDL-C) are routinely measured in clinical laboratories, whereas cholesterol in low-density lipoprotein (LDL-C) is calculated according to the Friedewald formula [8] or is directly measured. These conventional lipid biomarkers referred to as “Standard Lipid Panel” or “Traditional Serum Lipid Profile” are used in the clinical setting for the diagnosis of lipid disorders, monitoring the response of patients to lipid-lowering therapy, and also for the assessment of CVD risk. In addition, total cholesterol and HDL-C are used to calculate non-HDL-C (as total cholesterol minus HDL-C), which represents the cholesterol content in all apoB-containing lipoproteins and is used in most risk equations for estimating a 10-year atherosclerotic cardiovascular disease (ASCVD) risk score.

Since the landmark publications from the Framingham study [9], the aforementioned lipid biomarkers have been recognized as significant predictors of future CVD events, with lipid lowering as a well-established intervention to reduce CVD risk [2]. Despite that, the bulk of CVD events cannot be explained sufficiently by these serum lipid parameters, as well as by the well-recognized predisposing factors. Many patients suffer from CVD; however, they are presented with serum lipid levels within the recommended range. CVD remains the leading cause of mortality and morbidity worldwide, as the current preventive strategies are ineffective in a large proportion of the population.

Since human plasma consists of thousands of functionally and chemically diverse molecular lipid species [1], the aforementioned conventional lipids are poorly descriptive of the total serum lipidomic profile, and a wealth of metabolic information is currently underutilized in the clinical setting. Advances in the field of lipid analysis that occurred over the last decade could contribute to the identification and quantification of several different molecular serum lipid classes, as well as individual lipid species, including phospholipids, glycerophospholipids (phosphatidylcholine, phosphatidylserine, phosphatidylinositol, and phosphatidylethanolamine), sphingolipids (sphingomyelin), glycerolipids (triglycerides, diglycerides, and monoglycerides), sterols (esterified and free cholesterol), fatty acids (saturated and unsaturated fatty acids and poly- and monounsaturated fatty acids, as well as individual fatty acids such as linoleic acid, arachidonic acid, docosahexaenoic acid, and eicosapentaenoic acid), etc., consisting of the comprehensive lipid profiling beyond the traditional lipids measured by routine clinical chemical methods (Figure 1). Cholesterol and triglycerides are determined in laboratory routine; however, information on the fatty acids esterified in them is not provided. Given such a tight homeostatic regulation of lipid metabolism, the comprehensive characterization of serum lipidomic profile could unravel disturbances in lipid-related pathways occurring in clinically ‘silent’ and complex lipid-related diseases such as atherosclerosis, prediabetes, diabetes, CVD, etc. in the early prodromal stage when the first nonspecific disease symptoms occur to overt disease.

## 3. Lipidomics: A Holistic Approach for Advanced Research of Lipid Metabolism in Cardiometabolic Diseases

Lipidomics is considered to be a branch of metabolomics, to which it is very closely related and aims to the full characterization of lipid molecular species and of their biological roles with respect to expression of proteins involved in lipid metabolism and function, including gene regulation. The term lipidomics was used for the first time by Kishimoto et al. in 2001 [10]. Two years later, (Han and Gross) the definition for this new discipline were established [11], coinciding with the establishment of LIPID MAPS (www.lipidmaps.org, accessed on 1 December 2022). The International Lipid Classification and Nomenclature Committee under the sponsorship of the LIPID MAPS Consortium established a “Comprehensive Classification System for Lipids” based on well-defined chemical and biochemical principles together with ontology [12,13]. This classification system groups lipids into eight categories (fatty acyls, glycerolipids, glycerophospholipids, sphingolipids, sterols, prenols, saccharolipids, and polyketides), which are further divided into classes and subclasses based on the head group and type of linkages [14] (www.lipidmaps.org).

The last decade, there has been a major increase in interest in lipidomics, as biomedical research turns its attention to global profiling and precision medicine. Lipidomics is heavily relying on the principles and technologies of analytical chemistry for the analysis of lipid structures, quantification of discrete molecular lipid species, and interactions that collectively identify the dynamic changes of lipids during cellular perturbations. As such, all analytical procedures, strategies, and methodologies during lipidomic analysis should have a robust workflow. As depicted in Figure 2, a typical lipidomics workflow is composed of several experimental and analysis steps. Sample collection, storage, preparation, and lipid extraction methods are major critical steps since many biases may be introduced, potentially altering the lipid composition of the biological sample of interest [15]. Sample preparation methods mainly depend on the sample type (e.g., blood, urine, or saliva) and the analytical method being utilized (e.g., nuclear magnetic resonance (NMR) or mass spectrometry (MS)). Consistency of experimental methods (e.g., timing of collection, materials and reagents, and storage temperature) is paramount to enable acquisition of accurate and reproducible results. Next, one analytical technique (and sometimes several) is used for the detection and characterization of the compounds present in samples. Once data are acquired, raw peak intensity data are processed to permit further analysis. For example, in NMR, different steps precede identification of lipid molecules including spectral pre-processing consisting of noise reduction and baseline correction, spectral alignment, and spectra division into sections (i.e., bins) in untargeted omics analysis, followed by calculation of bin intensities and statistical tests to assign bins to a specific lipid compound. Data normalization, scaling, and transformation are also performed prior to data analysis and interpretation.

Regardless of the analytical approach chosen for the detection and characterization of lipid molecules, a broad range of statistical analyses can be performed to determine potential differences between “healthy” and diseased groups or between groups at different disease stages, including univariate (ANOVA, *t*-test, or nonparametric equivalents) and multivariate analysis, either in an unsupervised (principal component analysis (PCA)) or supervised (partial least square-discriminant analysis (PLS-DA)) manner. The last step of the lipidomics workflow is the data interpretation, the identification of lipid biomarkers and aberrant lipid-related metabolic pathways. For that, available software tools and databases including the Kyoto Encyclopedia of Genes and Genomes (KEGG), LIPID MAPS, and MetaboAnalyst, which further enable pathway and enrichment analysis, lipid mapping, and visualization, are widely used in applications of lipidomics [16]. In this review, attention will be focused on the comprehensive analysis of lipid composition of serum lipoproteins by proton NMR.

### 3.1. Sample Collection and Storage

The validity and the outcome of a lipidomic analysis can only be as good as the sampling process covering sample collection, preservation, and extraction. The biological matrices can be solid in nature (tissues or cells) or highly complex biofluids (plasma or serum, urine, amniotic or cerebrospinal fluid, etc.) and are collected from healthy individuals and/or patients (Figure 2). Each of these matrices has a unique compositional pattern consisting of proteins, metabolites, and lipids at substantially different concentrations.

The optimal protocol for the collection and storage of samples depends on multiple parameters, including the type of sample (urine, blood, extracts, or tissues), the physicochemical properties of the compounds of interest, etc. For example, the preparation procedure methods for solid samples are more labor intensive than those for biofluids because additional steps, such as sample homogenization, prior to lipid extraction are required. In contrast, those for biofluids tend to be more straightforward. The most clinical lipidomics studies are based on blood-derived samples (plasma/serum), which can be collected with low invasiveness and minimal health risk. Blood-derived samples contain lipids in the form of lipoproteins, as well as metabolites secreted by different tissues and cells in response to various physiological or pathological stimuli. As a consequence, serum and plasma can provide important information at a systemic level that is representative of the overall metabolic status of the human organism [17], which is strongly affected by health or diseased conditions, genetic variations, environmental factors, lifestyle, nutrition habits, and drugs. Below, the conditions of blood sample collection and storage will be discussed with a focus on lipoprotein isolation from serum samples, which is consistent with the scope of the present review article.

***Blood collection***: traditionally, blood is collected from a peripheral vein after an overnight fast (12–14 h). Serum is obtained after clotting by centrifugation, which allows the removal of fibrin clots, blood cells, and related coagulation factors, whereas plasma is obtained by adding anticoagulants (i.e., EDTA, citrate, and heparin) before removal of blood cells by centrifugation. Then, it is directly divided into prelabeled aliquots to limit freeze–thaw cycles and stored as fast as possible at −80 °C until the lipid extraction procedure is carried out. Blood collection tubes are a critical step in the preanalytical phase; however, only few data exist in the literature concerning their effect on the lipid components [18]. Because lipid levels can exhibit substantial circadian variations, the time point of blood collection should be kept consistent within a study [19]. In addition, detailed information concerning the fasting status of a participant, differences in temperature and time spent on the bench, before and after centrifugation, etc. [20], as well as visible signs of sample quality (i.e., hemolysis and lipemia) [21], should be recorded for later data interpretation.

***Blood-derived samples storage***: the storage conditions of blood-derived samples strongly affect the quality and quantity of lipidome. Many lipid components undergo chemical and enzymatic degradation [22]. For example, lipid oxidation may occur during collection, storage, and extraction and can affect lipid species that contain polyunsaturated fatty acid moieties [23]. Since the time that elapses between a sample’s collection and analysis may be long, several parameters including temperature, light, number of freeze–thaw cycles, etc. must be monitored to eliminate the degradation risk of lipid components. Because freeze–thaw cycles affect specific lipid classes, it is recommended that their number should be kept to a minimum, ideally less than two, and be constant within a study [24]. In addition, serum samples should be stored at −80 °C because the stability of lipids is considered sufficient for long-term storage and, therefore, for clinical lipidomics. Storage at higher temperatures (−20 °C, 4 °C) results in increased lipid degradation.

When the lipids carried on lipoprotein particles are within our interest, then serum lipoproteins including very-low-density lipoprotein (VLDL), intermediate-density lipoprotein (IDL), LDL, and HDL should be separated. Analytical methods such as ultracentrifugation, column chromatography, and gel electrophoresis can be used for their separation [25,26,27]. However, these methods are time consuming and not suitable for measuring large numbers of samples in clinical studies. In routine clinical laboratories, precipitation techniques are commonly used to separate atheroprotective HDLs from atherogenic apolipoprotein-B-containing or non-HDL lipoprotein particles (e.g., VLDL, IDL, lipoprotein (a) (Lp(a)), and LDL). With these techniques, the investigation of the lipid composition of HDLs, as well as the non-HDL fraction that expresses the total atherogenic lipid profile, is feasible. Briefly, a precipitant reagent containing polyanions and metal cation solutions is added to a sample (serum or plasma) and interacts with protein moieties of the lipoproteins selectively to form insoluble complexes with apo-B-containing lipoproteins that precipitate, whereas HDLs remain soluble in the supernatant [28].

### 3.2. Lipid-Extraction Techniques

Lipids are embedded in a biological matrix instead of being in a free form. Thus, prior to their detection, the structurally diverse lipid species must be extracted from their native state into suitable single-organic solvents or solvent mixtures. Lipid extraction is without a doubt the major limiting step to analyzing the complete set of lipids in biological systems in the lipidomics approach. For their extraction, we should use solvents that dissolve lipids readily and also overcome the interactions between lipids and components such as protein and polysaccharides via hydrophobic or van der Waals forces. If the biological matrices are tissues or cells, an additional homogenization step is required before the lipid extraction procedure. For the extraction of lipids, there are two significant challenges to overcome: extraction efficiency and complete removal of non-lipid constituents. Both challenges are influenced by parameters such as temperature, sample/solvent ratio, mixing times, and extraction under inert gasses, which in turn may be responsible for artifacts appeared in the identification of lipid molecules and the inconsistencies in their quantification.

A variety of well-established lipid-extraction techniques have been reported in the literature including monophasic (i.e., single organic solvent extraction) and biphasic approaches (i.e., liquid–liquid extraction (LLE) and solid-phase extraction (SPE)) [29,30,31]. Monophasic lipid extraction methods have been developed to promote full recovery of a broader range of lipid classes, facilitating a more comprehensive lipidomic analysis. Simultaneously, these methods may also limit the loss of more polar lipid species to partial or full partitioning to the aqueous phase during the biphasic methods, thus facilitating their recovery [32]. On the other hand, LLE approaches represent the most predominant commonly employed approach in lipidomic studies and can entail a broad array of organic solvents such as chloroform, methanol, methyl-tert-butyl ether (MTBE), isopropanol, and butanol [33,34]. One issue presented by these methods is the high volatility of chloroform solvent, as evaporation during handling is a potential source of experimental error, sometimes avoided by additional evaporation/reconstitution steps. In addition, the collection of the lower organic phase in two-phase liquid–liquid extractions can also be cumbersome. Note that lipids of extremely low endogenous abundance (minor lipids) may require enrichment via solid-phase extraction.

The aforementioned techniques are reasonably efficient in terms of conferring a general lipidome representative of the vast diversity of endogenous lipids; however, the choice of a particular lipid-extraction protocol and organic solvent depends predominantly on the lipids of interest, the range of concentrations at which they exist, and the samples being analyzed. According to the scope of the present review article, we will focus on the methods used for extraction of all classes of detectable lipids from serum lipoproteins with minimal contamination of non-lipid molecules such as proteins and carbohydrates. LLE chloroform/methanol-based protocols, such as Folch [35] or Bligh and Dyer [36], have been used since the late 1950s with few modifications and are considered the gold-standard two-phase methods for the extraction of lipids from each lipoprotein fraction. Generally, the Folch method is preferred to extract lipids from solid tissue samples, whereas that of Bligh and Dyer is considered advantageous for biological fluids [37]. Both two-phase extraction approaches partition lipids into an organic phase by using chloroform/methanol solvents in different proportions; in the Folch method, the ratio of chloroform:methanol is 2:1 (*v*/*v*), whereas in Bligh and Dyer, the chloroform:methanol ratio is 1:2 (*v*/*v*).

Following lipid extraction, the stability of the extracted lipids is also an important consideration, particularly if lipid molecules contain unsaturated double bonds that can be subjected to oxidation in the presence of oxygen in air. To minimize this effect, the removal of air from the headspace of sample vials containing the lipid extracts, by passing inert gases, such as nitrogen, over the mouths of the vials prior to sealing them, can also help minimize the occurrence of lipid oxidation reactions.

### 3.3. Analytical Techniques for Lipidomics Analysis

After the lipid extraction procedure, the next key step in the lipidomic workflow is the qualitative and quantitative analysis of lipid classes extracted from biological samples. Because lipids are highly complex and structurally diverse with heterogeneous physiochemical properties their comprehensive analysis in a direct and quantitative manner presents considerable analytical challenges.

Historically, the earliest attempts for the study of lipid composition of cells, body fluids, and tissues were performed by conventional chromatographic separation-based techniques such as high-performance liquid chromatography (HPLC), thin-layer chromatography (TLC), gas chromatography (GC), liquid chromatography (LC), or electrophoretic techniques such as capillary electrophoresis (CE) [38,39,40]. Although these traditional methods provide the simultaneous detection and quantification of several lipids in a given sample, they are, however, very tedious and time-consuming as usually more than a single step is needed and specific preselection of experimental conditions is required. Thus, these techniques could not meet the requirements for the application of lipidomic approaches to the study of human pathological states because they were possibly plagued by cumulative errors. In recent years, various techniques, such as TLC, GC, LC, and CE, are usually coupled with MS as a detector for structural determination and quantitative analysis of lipids because lipid signals in complex biological matrices tend to be overlapped if there is no prior analytical separation step [41,42]. The chromatography-MS hyphenated methods can obtain comprehensive information on almost all lipids in a sample, while chromatography separation may cost more time than direct-infusion MS. 

The two most commonly used analytical tools for lipid analyses are the NMR and MS techniques. MS is a powerful analytical technique for the detection of various classes, subclasses, and individual molecular species of lipids in a biological sample with high sensitivity and high throughput [43,44,45]. Applications of MS in comprehensive lipid analysis have been described previously [46,47,48]. Compared to MS, NMR spectroscopy also presents a powerful fingerprinting analytical method for the qualitative and quantitative characterization of targeted lipid molecules or in complex mixtures in biological samples. It requires minimal and unsophisticated sample preparation and does not induce degradation of the sample upon analysis. In addition, it is a highly reproducible and directly quantitative technique in spite of its lower sensitivity (overlapping signals in either ^1^H NMR or ^31^P NMR) in the high micromolar to millimolar range compared with MS-based techniques and the low natural abundance of ^13^C for ^13^C NMR. Its sensitivity can be improved by using very strong superconducting magnets and cryoprobes that operate the receiver circuitry at low temperatures to reduce electronic noise. In addition, NMR has been successfully hyphenated to LC as liquid chromatography-NMR (LC-NMR) for accurate lipid separation and subsequent structural elucidation tackling the problem of signal overlaps.

A multidimensional NMR technique could obtain a better resolution of the signals attributed to lipids without sacrificing the sensitivity, as it allows their dispersion into two dimensions, providing an information-rich two-dimensional lipid map in which key unique peaks can be resolved and used for the identification and quantification of specific lipid molecules [49]. The development of the two-dimensional NMR techniques may bring new vitality for NMR in lipidomics analysis [41]; however, are not yet widely utilized in the field of clinical lipidomics. This is mainly due to the long-time experiments involved—up to several hours of acquisition—which do not meet the high-throughput requirements associated with omics approaches. Recently, two-dimensional NMR approaches such as ultrafast (UF) NMR and nonuniform sampling (NUS) have emerged to tackle the issue of acquisition time and have already shown great potential and usefulness in the field of omics area [50], either used for fingerprinting [51] or associated with a calibration procedure for targeted analyses when absolute quantification is necessary. 

In the field of serum lipoprotein research, NMR has been proven to be a main tool in the study of the structure of lipoprotein particles. The first publication on lipoprotein structure was entitled “Structure of human serum lipoproteins: nuclear magnetic resonance supports a micellar model”, by Steim et al., involved a magnetic field frequency of 60 MHz and was dated from 1968 [52]. Since then, over a thousand publications have been cited in the term “NMR and lipoprotein” in the PubMed search. Most of them concern the field of basic research of lipoprotein structure and function, and significant numbers of them concern the assessment of the detailed determination of lipoprotein subclass particles’ distribution in plasma. The NMR signal emitted by each class of lipoproteins, VLDL, LDL, and HDL has a unique spectral line shape that makes it possible to differentiate one from the other. Furthermore, accurate quantification of particle numbers occurs because the amplitudes of the individual subclass NMR signals are directly proportional to the number of particles emitting the signal. This NMR-based lipoprotein subclasses analysis offers several significant advantages over the other analytical procedures since it is very fast, measures simultaneously all of the lipoprotein subclass concentrations, does not use reagents, completely eliminates the need of physical separation, and requires minimal sample manipulation. This approach that received wide clinical interest is mainly focused on determining the risk for CVD [53,54,55].

#### NMR-Based Analysis of Lipid Extracts of Serum Lipoproteins

Historically, the first application of ^1^H NMR spectroscopy in the detailed analysis of lipids extracted from cell membranes was pioneered by Sparling et al. in 1989 [56]. Then, several studies have been published, mainly on the lipid extracts from cell membranes and tissues [57,58,59,60,61], providing enough knowledge concerning experimental protocols and methodological aspects for lipid extraction and assignment of lipid classes, e.g., phospholipids (glycerophospholipids and sphingolipids), cholesterol (free and esterified), triglycerides, and fatty acids.

In this part of the review, the application of the NMR on the comprehensive analysis of lipids extracted from serum lipoproteins will be discussed. It is well-known that the lipids located on the core and surface of serum lipoproteins are qualitatively similar but quantitatively different. For example, triglycerides are the predominant core lipid in VLDL, while cholesterol esters predominate in the cores of HDL and LDL particles. Compared to LDL, HDL contains higher amounts of phospholipids, particularly phosphatidylcholine. Regardless of the lipoprotein class, the lipids will appear in the NMR spectrum with more than one signal, and their intensity is linearly related to their concentrations. For example, the signal attributed to triglycerides will appear in the NMR spectra of both LDL and HDL lipid extracts but with higher intensity in the NMR spectrum of LDL lipid extracts compared to that of HDL because of the higher content of triglycerides in the LDL particle’s core.

A typical ^1^H NMR spectrum of the total lipid extract of HDL lipoprotein particles is shown in Figure 3. Each lipid molecule appears in the spectrum with more than one signal; its chemical shift depends on the chemically different proton groups in its structure. Certain signals that have a similar chemical nature with signals from other molecules are superimposed, whereas others appear well-resolved, having a unique chemical shift, and are therefore suitable for identification and quantification. As seen in Figure 3 and Table 1, lipid molecules located in the surface and cores of HDLs can be identified and quantified from characteristic well-resolved signals in the NMR fingerprinting including cholesterol (total, free, and esterified), triglycerides, phospholipids, and fatty acids. In addition, a more detailed phospholipid profile (phosphatidylcholine, lysophosphatidylcholine, phosphatidylethanolamine, phosphatidylinositol, ether glycerolipids, plasmalogens, sphingolipids, and sphingomyelin) and fatty acid pattern (unsaturated and polyunsaturated, arachidonic and eicosapentaenoic acid, and docosahexaenoic and linoleic acid) can also be determined.

### 3.4. Types of Lipidomic Approaches

Regardless of the analytical technique used for the lipidomic analysis, the study and interpretation of the resulting lipidomic data can be accomplished using either a targeted approach or an untargeted approach [62].

In a targeted approach, the physicochemical properties of a pre-defined set of lipid constituents of interest are a priori known. This allows for a more selective biological sample preparation optimized for the set of lipids analyzed, thus providing their absolute concentrations. As opposed to the targeted strategy, the untargeted lipidomic approach aims to confer a general overview and the reproducible measurement of as many lipid molecules as possible in the total lipid extract of a given biological sample. Thus, a global lipidome is yielded where alterations in lipid patterns occurring under biological perturbation or stimulation can be detected, or aberrant lipid compounds, whose structure and chemical formulas are unknown, can be reliably and unambiguously identified. With this approach, peak areas are reported for lipid compositional characteristics instead of absolute concentrations, resulting in semi-quantitative lipidomic data.

## 4. Disturbances in Lipidome of Serum Lipoproteins in Cardiometabolic Diseases

Lipoprotein metabolism is an extremely complex process during which lipoprotein particles exchange their proteins and lipids and undergo extensive remodeling by lipid transfer proteins, lipoprotein receptors, and lipophilic enzymes. Imbalances in lipid metabolism are linked to various cardiometabolic diseases such as obesity, diabetes, chronic renal failure, or cardiovascular disease. To understand the pathogenesis of these diseases and subsequently develop successful treatment, it is necessary to be able to track the aberrant lipid-related metabolic pathways. These states may contribute to the vulnerability of serum lipoproteins to compositional and structural modifications, causing disproportionation in two or more of their components beyond cholesterol and triglycerides, which are routinely determined.

Before investigating the alterations occurring in serum lipoproteins in cardiometabolic diseases, it is important to define the lipid composition among healthy individuals. Studies have shown that the total lipid content differs across lipoproteins [64,65,66]. VLDLs are mainly composed of triglycerides and LDLs mainly of cholesterol esters, while HDLs are primarily composed of cholesterol esters and phospholipids. Compared to LDLs, HDLs contain higher percentages of lyso-phosphatidylcholine. Lipoprotein particles also differ in individual lipid classes; for example, lyso-phosphatidylcholine is predominantly contained in HDLs, whereas phosphatidylcholine is mainly contained in HDLs and LDLs. The fatty acid content of phospholipids, triacylglycerols, and cholesterol seems comparable across HDLs, LDLs, and VLDLs. However, polyunsaturated fatty acids esterify lipids in HDLs and LDLs, while monounsaturated and saturated fatty acids are more likely to be distributed across VLDLs and chylomicrons.

HDL or atheroprotective lipoproteins, apart from the well-recognized antiatherogenic role of HDLs in reverse cholesterol transport (RCT), have been postulated to exert numerous functions including antioxidant, anti-inflammatory, vasodilatory, antithrombotic, immunomodulatory, and endothelial repair properties that may contribute to the protection from atherosclerosis. Structure–function studies suggest that the ability of HDLs to exert these functional properties is inextricably related to the overall structure and composition (lipidome or proteome) [67], rather than cholesterol content alone. As can be expected, remodeling of HDLs can affect their stability and structure, and therefore the eventual functional capability.

The influence of disproportions in lipid components on the HDL structure has been investigated in reconstituted discoidal and spherical HDL particles. Changes in core neutral lipids (cholesterol esters and triglycerides) have distinct effects on an HDL particle’s stability and induce conformational and structural changes on the primary apoprotein, apoAI. An increase in triglyceride content has a negative effect on a particle’s stability, whereas the stability is enhanced by the presence of cholesterol ester molecules [68]. The conformation of apoAI appears to be extremely sensitive to the neutral lipid content. Both cholesterol esters and triglycerides reduce the α-helix content of apoAI, but while cholesterol esters increase the α-helix stability, triglycerides reduce α-helix stability, increasing the propensity of apoAI to dissociate from HDL and, therefore, to be cleared from the plasma [69]. Thus, a low cholesterol esters/triglycerides ratio in the particles’ cores decreases the stability of LpA-I particles and give rise to specific changes in the conformation and charge of apoAI. In addition, triglyceride-rich HDLs presented abnormality in cholesterol efflux, esterification, and transport, which may contribute to and/or enhance the atherosclerosis process [70]. This lipid modification also enhances in vivo the metabolic clearance of HDL apoAI [71] and impairs its anti-inflammatory capacity with a dose-dependent manner.

Phospholipids play an important role in regulating the structure and function of HDLs. Rye KA et al. in 2000 showed that phospholipids depletion not only had a major effect on the size of HDLs but also influenced the cholesterol ester transfer protein (CETP)-mediated remodeling and dissociation of apoAI [72]. The fluidity of the phospholipids surface monolayer in HDLs is strongly determined by the amount and proportions of phospholipids [73] and is particularly important for both scavenger receptor class B member 1 (SR-BI)- and ABCA1-mediated cholesterol efflux [73]. Enrichment of HDLs with phosphatidylcholine and sphingomyelin enhances the bidirectional flux of cholesterol from SR-BI expressing cells to HDL by two different mechanisms: phosphatidylcholine increases cholesterol efflux, while sphingomyelin decreases the influx of cholesterol [74]. Phospholipids are also responsible for the ability of HDLs to inhibit the cytokine-mediated increase in the endothelial cell expression of adhesion molecules [75], reducing the recruitment of blood monocytes into the arterial wall, a process known as anti-inflammatory activity. Phosphatidylcholine was thought to be the main lipid component responsible for this activity of HDLs [75].

As mentioned above, disturbances in different HDL lipid molecules beyond cholesterol could lead to the formation of dysfunctional or even pro-atherogenic particles. The lipid composition of HDLs appears to be compromised by acute and chronic inflammatory conditions, as well as by metabolic decompensation, including obesity, metabolic syndrome, insulin-resistance states, nonalcoholic fatty liver disease (NAFLD), chronic renal failure, cardiovascular disease, etc. This section reviews the impact of cardiometabolic diseases on the modification of HDL lipidome that compromise the functionality of HDL.

### 4.1. Obesity and Metabolic Syndrome (MetS)

The prevalence of overweight and obesity is increasing and represents a primary health concern because of the increased risk for development of T2D and CVD. The specific underlying mechanisms linking the expansion of adipose tissue, a characteristic of the obese state, to cardiometabolic comorbidities associated with obesity are still poorly understood. Serum HDL-C levels and cholesterol efflux capacity are often decreased, especially when obesity is associated with metabolic syndrome (MetS). The latter includes a clustering of clinical findings of abdominal obesity, hyperglycemia, hypertriglyceridemia, hypertension, and low HDL-C levels. MetS is tightly associated with lipid disorder, which relies on the ectopic accumulation of lipids (lipotoxicity), as well as on mitochondrial dysfunction leading to an increased CVD risk. In addition, it is characterized by several metabolite changes in plasma, reflecting abnormalities in several metabolic pathways. However, the specific underlying pathophysiologic mechanisms pertaining to how altered lipid metabolism can potentially result in disease onset, have largely remained unaccounted for.

Central obesity is associated with modifications of the atherogenic nature to lipoprotein distribution and composition including increased synthesis of larger VLDL particles and a shift to small, dense LDLs that have been positively associated with increased CVD risk [76]. Besides the low levels of cholesterol content in HDL lipoproteins, several alterations occurring in other lipid constituents closely affect the quality and functionality of HDLs. Chronic inflammatory milieu, in which obese individuals are exposed, impacts the antiatherogenic properties of HDLs [77]. Several enzymes, such as lecithin-cholesterol acyltransferase (LCAT) and CETP that are carried by HDLs and are involved in their remodeling and metabolism, have an altered pattern of activities [78]. Increased CETP activity observed in obese individuals leads to alterations in HDL core lipids. Accumulation of triglycerides inside HDLs’ core paired with depletion of cholesterol esters have been observed, resulting in an increased triglycerides/cholesterol esters ratio and altered core fluidity and functionality [78,79]. The enrichment of HDLs in triglycerides also promotes conformational changes of apoAI and its detachment from particles, reducing their ability to inhibit lipoprotein oxidation and, thus, increasing oxidative stress [80]. Alterations in surface lipid constituents have also been observed. For example, the amounts of free cholesterol and total and individual phospholipids are profoundly altered [79], leading to the modification of quality and the functional impairment of HDLs and to an increased CVD risk in obese individuals.

HDL isolated from patients with MetS display a reduced ability to activate eNOS and, consequently, to promote the production of nitric oxide mainly because of depletion of HDLs in a sphingolipid named sphingosine-1-phosphate (S1P) [81,82]. The impaired ability of HDLs to promote nitric oxide production could be responsible for the endothelial dysfunction commonly detected in these patients. Moreover, enrichment in triglycerides, depletion and modification of apoAI, and increased HDL surface rigidity caused by altered phospholipid profile reduce the antioxidant capacity of HDLs [83,84,85] and, thus, their ability to accept and inactivate oxidized phospholipids from LDL [86].

### 4.2. Insulin Resistance-Type 2 Diabetes-Prediabetes

According to a WHO report, in 1980–2014, the number of people suffering from diabetes increased from 108 to 422 million. The World Diabetes Federation predicts that the incidence will reach 700 million by 2045. Disturbances in glucose homeostasis and lipid metabolism, both well-recognized predisposing factors for CVD, are highly interrelated at multiple metabolic crossroads. Thus, aberrations in pathways involved in lipoprotein metabolism could at least partially explain the risk for the development of atherosclerosis in individuals with carbohydrate-metabolism derangement.

In insulin-resistance states, HDLs undergo several structural and compositional disturbances that are beyond their cholesterol content. As a result, functionally deficient HDL particles are formed by several proposed mechanisms [87]. For example, increased content of oxidized lipids and advanced glycation end products (AGE) have been found in T2D because of the presence of systemic oxidative and glycemic stress [88], whereas apoAI and S1P concentrations are decreased by their displacement by other proteins such as serum amyloid A (*SAA*) [89] because of underlying chronic inflammation [90].

Meanwhile, the functional status of HDLs is strictly related to alterations occurring in lipid components. Increased triglyceride content in HDL core caused by insulin resistance and elevated CETP activity [91,92], abnormal fatty acid pattern, and a phospholipid profile [91] that confers greater rigidity to a particle’s surface have been observed in T2D patients. Since phospholipids are a major determinant of the ability of HDL to accept cholesterol from cells, the low phospholipid levels could explain the impairment of the acceptor capacity and the reduced cholesterol efflux of HDL in T2D [93], possibly because of the impairment of both ABCA1- and SR-BI-mediated pathways and LCAT activity [94]. In addition, the anti-inflammatory function of HDLs, measured as the inhibition of cell adhesion molecule or cytokine expression by both endothelial cells and monocytes, is also impaired. Although the responsible mechanisms have not been fully elucidated yet, it has been suggested that the low phosphatidylcholine levels herein contained and the glycation or oxidation of HDL-associated proteins could play vital roles.

Prediabetes is a clinically silent, insulin-resistant state that often remains undiagnosed and is associated with an increased risk for the development of T2D [95] and CVD [96]. Although multiple factors have been involved in the pathogenesis of prediabetes, the pathophysiologic mechanisms underlying the progression of prediabetes to overt T2D and also the proneness of these patients to developing premature atherosclerosis remain obscure. Concerning the alterations occurring in HDLs in patients with prediabetes, few data exist in the literature. Recently, an NMR-based lipidomics analysis conducted by our group revealed that the HDLs isolated with prediabetes presented subtle but statistically significant atherogenic alterations in the surface and core particle’s lipidome compared to the normoglycemic group, notwithstanding that conventional serum lipid parameters were within the normal range. In addition, these changes were qualitatively similar but quantitatively milder compared to those observed in T2D patients [91]. It is worth noting that the observed alterations in the HDL lipidome occurred despite HDL-C levels having been selected to be statistically similar among the normoglycemic group, prediabetes and diabetes patients.

### 4.3. Non-Alcoholic Fatty Liver Disease (NAFLD)

In addition to adipose tissue, the liver is another key organ early associated with insulin resistance and diabetes. NAFLD is one of the most common forms of chronic liver diseases in the Western countries, affecting approximately 25% of the general population, and is a major risk factor leading to MetS, T2D, CVD, chronic liver disease, and liver failure [97]. It is characterized by the hepatic intracellular accumulation of lipids, mainly triacylglycerols, and the subsequent formation of lipid droplets in hepatocytes [98].

Although there has been remarkable progress in the elucidation of NAFLD pathogenesis, the pathophysiological pathways underlying lipotoxicity and the transition of simple steatosis to NASH are still incompletely understood [99]. Liver biopsy remains the only reliable but invasive method to diagnose NAFLD and differentiate NASH from simple steatosis. Since the deposition of lipids in the liver is the major hallmark feature of NAFLD, the determination of specific types and amounts of lipids secreted from the liver in the form of lipoprotein particles should generate a characteristic lipid signature for the presence and progression of NAFLD. Alterations occurring in lipid molecules identified by lipidomics approaches may have utility as non-invasive lipid biomarkers of the presence and progression of disease [100]. Lipidomic studies revealed marked changes in the fatty acid pattern and phospholipid composition in liver samples of NAFLD patients, suggesting that perturbations in lipid metabolism are a key factor in the pathogenesis and progression of NAFLD [101,102]. Alterations in the lipidome of serum samples and of lipoprotein particles [103] could closely reflect those occurring in a liver lipidome and affect future therapeutic strategies, but limited data exist in the literature [104,105].

### 4.4. Chronic Renal Failure

Chronic renal failure is associated with an increased risk of CVD as these patients develop accelerated atherosclerosis [106]. Renal dysfunction is associated with many perturbations in lipoprotein metabolism leading to dyslipidemia and accumulation of atherogenic lipoprotein particles.

Studies have shown that oxidized LDL-C levels increase as kidney function declines [107]. The chronic renal-failure-associated dyslipidemia is characterized by elevated levels of TGs, LDL-C, accumulation of apoB-containing lipoproteins, and low HDL-C levels [108]. In addition, HDL metabolism is impaired and HDL-3 particles are not maturated into HDL-2 because of LCAT deficiency [109]. HDLs isolated from patients with chronic renal disease are less able to activate eNOS and their capacity to repair endothelium is also compromised. The differences in HDL properties may be attributed to changes in the HDL-associated lipidome; such modifications involve alterations in the amount and type of lipids bound to the HDLs. However, limited data exist in the literature concerning the alterations occurring in HDL lipid composition in these patients. Honda et al. [110] showed decreasing levels of HDL3 and ApoAI in the HDL3 subfraction along with the worsening of chronic renal-failure severity. Tolle et al. [111] demonstrated that HDL isolated from patients with chronic kidney disease was enriched in SAA and exhibited decreased anti-inflammatory capacity as assessed on the basis of its ability to inhibit monocyte chemoattractant protein-1 formation in vascular smooth muscle cells.

### 4.5. Cardiovascular Disease

CVD is mostly associated with alterations of blood levels for one or more lipid classes. The cholesterol content of HDLs is not a causal marker because many functional properties of HDL are exerted either by entire particles or specific constituents (lipid or protein) other than cholesterol. It has been shown that overt CVD dramatically alters the composition and overall structure of HDLs and endows the particles with abnormal biological and physico-chemical properties, and finally alters quality and impairs functional status. Indeed, the inflammation burden of CVD induces changes in the protein or lipid components of HDLs, beyond cholesterol content, resulting in the formation of dysfunctional particles [112]. However, the exact pathophysiological mechanisms underlying these disturbances, which may accelerate rather than reduce future risk of major adverse events, have not been fully understood.

HDLs lose their atheroprotective properties and become gradually proatherogenic. Cholesterol efflux capacity is decreased [113], while numerous endothelial protective functions appear to be impaired. Reduced nitric oxide bioavailability is evident in HDLs isolated from patients with established coronary heart disease (CHD) [114], whereas HDLs have been shown to promote rather than protect against LDL oxidation [115]. Moreover, an increased expression of inflammatory adhesion markers was observed [115].

The first studies conducted in HDLs isolated from CVD patients revealed alterations in individual lipid molecules and specifically in phospholipids, whose content was lower in these patients [116,117,118,119]. Interestingly, the severity of the disease was more strongly correlated with the decrease in HDL-phospholipids than in HDL-C. Of note, low HDL-sphingomyelin levels have the strongest association with the presence of CHD among all HDL-related parameters studied and is the only one with a significant and independent correlation with the number of coronary stenoses [118]. In recent years, the emerging lipidomic approaches allowed for the high-throughput profiling of many lipid components simultaneously, contributing to an in-depth understanding of the dysregulation of lipid metabolism and exploring altered lipid-related metabolic pathways underlying CHD. NMR-based lipidomic studies revealed that the onset and, interestingly, the progression of CHD is related to modifications in many surface and core lipid components of HDL (atheroprotective) and non-HDL (atherogenic) lipoproteins, beyond cholesterol content, as well as in fatty acids esterified in these lipids [28,120], indicating that the detailed investigation of lipid composition of lipoproteins could reveal aberrant lipid-related pathways that could not be determined with the traditional serum lipid parameters.

## 5. Conclusions

The increasing global burden of cardiometabolic diseases highlights the urgent clinical need for better personalized prediction and intervention strategies for the right individual at the right time (at an early stage). The knowledge of the alterations in lipid metabolic pathways in a broad spectrum of CVD etiologies could offer us valuable information on how to control lipid metabolism. To this aim, the advent of lipidomics approaches and other omics approaches have led to tremendous progress in the cardiometabolic diseases research field in last two decades. Integrating the emerging multiomics with lipidomics holds a great potential in moving toward this goal.

## Figures and Tables

**Figure 1 diagnostics-13-00721-f001:**
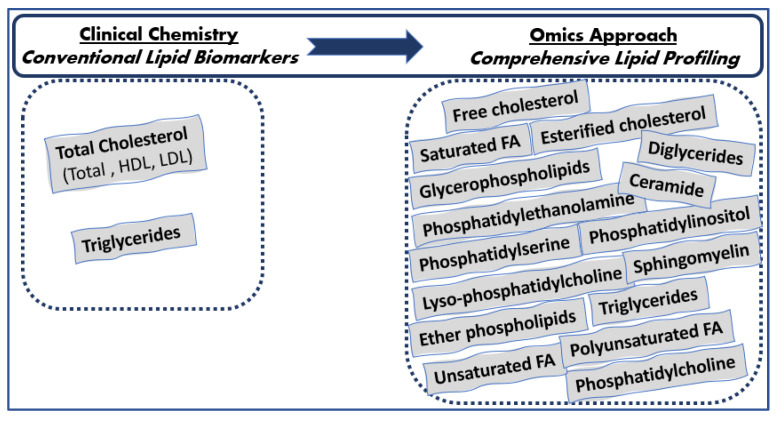
Lipids determined in everyday clinical laboratories vs. lipids determined with the lipidomics approach.

**Figure 2 diagnostics-13-00721-f002:**
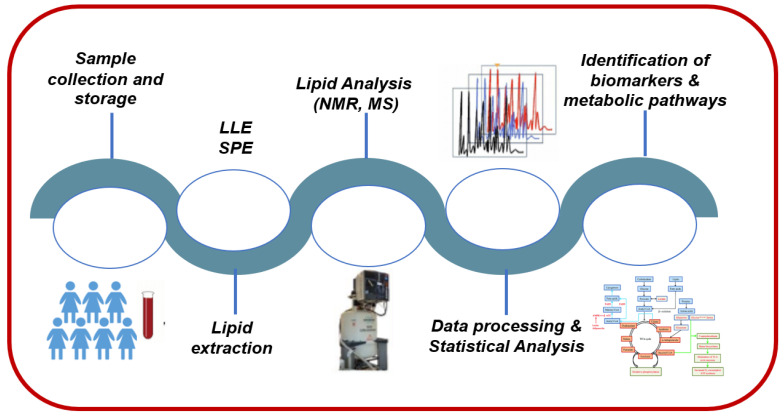
A representative lipidomics workflow.

**Figure 3 diagnostics-13-00721-f003:**
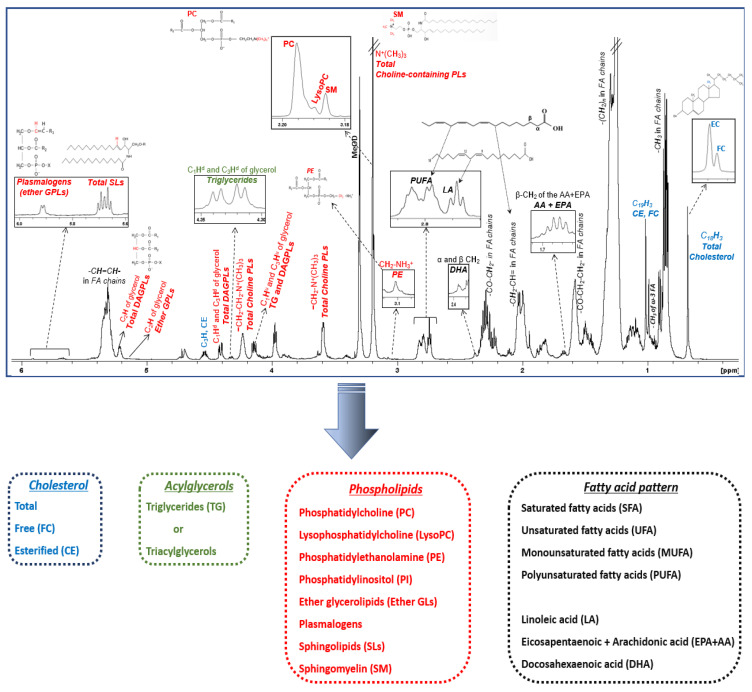
^1^H NMR spectrum of an HDL lipid extract. Figure is adopted from [63]. Peak assignments are summarized in Table 1.

**Table 1 diagnostics-13-00721-t001:** Protons and chemical shift (in ppm) for the lipid constituents and headgroups identified in HDL lipid extract by NMR and selected signals for lipid quantification.

Lipid Constituents and Headgroups	^1^H NMR Signal Assignment	Chemical Shift (ppm)	Quantification of Lipids from Selected Well-Resolved NMR Signals
**Cholesterol molecule**	C_18_***H***_3_	0.68	Total Cholesterol, FC, CE
	C_26_***H***_3_, C_27_***H***_3_, C_21_***H***	0.87	
	C_19_***H***_3_	1.00	
	C_3_***H***	3.40	
	C_6_***H***	5.36	
**Glycerol backbone**	C_1_***H***u and C_3_***H***u of glycerol backbone of TG and DAGPLs	4.16	
	C_1_***H***d and C_3_***H***d of glycerol backbone of TG	4.32	TG
	C_1_***H***d and C_3_***H***d of glycerol backbone of DAGPLs	4.40	
	C_2_***H*** of glycerol backbone in ether glycerophospholipids	5.15	Ether GPLs
	C_2_***H*** of glycerol backbone in total DAGPLs	5.18	Total diacyl glycerophospholipids (DAGPLs)
	C_2_***H*** of glycerol backbone in TG	5.22	
**Sphingosine moiety**	-CH_2_-CH=C***H***CHOH	5.40	
	-CH_2_-C***H***=CHCHOH	5.70	Total SLs
**Head-group and substituent**	-CH_2_-CH_2_-N^+^(C***H***_3_)_3_ (choline)	3.20	Total choline-containing PLs (PC, SM, LPC)
	-CH_2_-C***H***_2_-N^+^(CH_3_)_3_	3.59	
	-C***H***_2_-CH_2_-N^+^(CH_3_)_3_	4.24	
	-CH_2_-C***H***_2_-NH_3_^+^ (ethanolamine)	3.10	PE
	-OC***H***=CHCH_2_	5.90	PLA (ether GPLs)
**Fatty acid chains**	ω-C***H***_3_ (methyl) in fatty acyl chains	0.88	
	ω-C***H***_3_ (methyl) of total omega-3 FA	0.95	
	-(C***H***_2_)n- (methylene) in fatty acyl chains	1.30	
	-CO-C***H***_2_-CH_2_- (β-methylene) in the fatty acyl chains	1.59	
	β-C***H***_2_ (β-methylene) of the sum of AA+EPA	1.67	AA (20:4 ω-6) + EPA (20:5 ω-3)
	-C***H***_2_-CH=(allylic) in fatty acyl chains	2.04	UFA
	-CO-C***H***_2_ (α-methylene) in the fatty acyl chains	2.30	Total FA
	α and β C***H***_2_ (methylene) of DHA	2.38	DHA (22:6 ω-3)
	-CH=CH-C***H***_2_-CH=CH- of linoleic acid	2.75	LA (18:2 ω-6)
	-(CH=CH-C***H***_2_-CH=CH)n, n > 1 in the fatty acyl chains	2.80	PUFA
	-C***H***=C***H***- in the fatty acyl chains	5.36	

**Key:** AA, arachidonic acid; CE, cholesteryl ester; DAGPLs, diacyl glycerophospholipids; DHA, docosahexaenoic acid; EPA, eicosapentaenoic acid; FA, fatty acids; FC, free cholesterol; GPLs, glycerophospholipids; LA, linoleic acid; LPC, lysophosphatidylcholine; PC, phosphatidylcholine; PE, phosphatidylethanolamine; PLA, plasmalogens; PUFA, polyunsaturated fatty acids; SLs, sphingolipids; SM, sphingomyelin; TG, triglycerides; UFA, unsaturated fatty acids.

## Data Availability

No research data available.

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
