# Peer review of "Expanding the Molecular Disturbances of Lipoproteins in Cardiometabolic Diseases: Lessons from Lipidomics"

_diagnostics, 2023, doi:10.3390/diagnostics13040721_

Round 1

Reviewer 1 Report

This is a very nice overview of the application of lipidomics in the study of serum lipoproteins in cardiometabolic disease.   

What is the main question addressed by the research? This is not a research study; this is just a comprehensive overview of the application Lipidomics in the study of serum lipoproteins in cardio-metabolic disease. That’s why for performing this literature review, the author did not get any external fund. In this review, the author extracted useful and related materials for lipoproteins through literature review of about 114 references mostly from 20 years ago to present. The essence of this article is to improve the knowledge of the alterations in lipid metabolic pathways in a broad spectrum of CVD etiologies. This pathway could offer us valuable information how to control lipid metabolism.

Is it relevant and interesting? Yes, the extracted materials are relevant and interesting particularly for medical students, research fellows, and also for clinician.

How original is the topic? We can find all these materials using different sources (e.g. books, papers, lectures, etc.). However, the art of the author was to collect all these information in a package and summarized them in a good format in the 4 sections. So, this part is original not the topic.

What does it add to the subject area compared with other published material?  

Is the paper is well written? The format of the paper was well-organized and easy to follow, but the writing itself could be improved in terms of clarity and concision in some sections such as 4.2 for example the differences between pre-diabetes and diabetes is not clear or the role of Lipidomics in development of insulin resistance needs to be more explained.

Is the text clear and easy to read? Yes, in my opinion the text is easy to read.

Are the conclusions consistent with the evidence and arguments presented? Yes, in general. However, I would like to see the challenges that are exist for the Lipidomics research studies at the present.

Do they address the main question posed? The main question that the author tried to address in the beginning of the paper was the shifts from traditional serum lipid measurements that are poorly descriptive of the total serum lipidomic profile to comprehensive lipid profiling is an urgent need for early diagnosing the metabolic syndromes and prevent cardio-metabolic disease. In my opinion the methodology and conclusion section of the paper are aligned with the main research question and it provides a convincing answer to the question that was posed.

Author Response

Reviewer 1

I thank the reviewer for the constructive comments. Please see my answers in red.

This is a very nice overview of the application of lipidomics in the study of serum lipoproteins in cardiometabolic disease.   

What is the main question addressed by the research? This is not a research study; this is just a comprehensive overview of the application Lipidomics in the study of serum lipoproteins in cardio-metabolic disease. That’s why for performing this literature review, the author did not get any external fund. In this review, the author extracted useful and related materials for lipoproteins through literature review of about 114 references mostly from 20 years ago to present. The essence of this article is to improve the knowledge of the alterations in lipid metabolic pathways in a broad spectrum of CVD etiologies. This pathway could offer us valuable information how to control lipid metabolism.

I thank the reviewer for the nice comment.

Is it relevant and interesting? Yes, the extracted materials are relevant and interesting particularly for medical students, research fellows, and also for clinician.

I thank the reviewer for the nice comment.

How original is the topic? We can find all these materials using different sources (e.g. books, papers, lectures, etc.). However, the art of the author was to collect all these information in a package and summarized them in a good format in the 4 sections. So, this part is original not the topic.

I thank the reviewer for the nice comment.

What does it add to the subject area compared with other published material?  

Very interesting point. Searching in the PubMed, many reviews concerning the role of serum lipidome in pathological conditions have been published.

The present review manuscript focuses on the role of lipidome of serum lipoproteins in cardiometabolic diseases using omics approaches. A detailed lipidomics workflow especially for the analysis of lipidome of serum lipoproteins is presented, providing useful information on the experimental steps; from sample collection to clinical interpretation of the robust  lipidomic data received from the omics analysis.

Almost 10 years before, Kontush A et al. published an article entitled “Lipidomics as a tool for the study of lipoprotein metabolism.” The authors have emphasized on the alterations of serum lipidome but not of lipoproteins’ lipidome (which is presented in our review) in pathological conditions, the effects of lifestyle and Dietary Modifications and of Lipid-Modifying Treatments on that. No lipidomics workflow was included in that study. Recently, Ding M et al. also published a review article entitled “A Review of Lipidomics of Cardiovascular Disease Highlights the Importance of Isolating Lipoproteins.” The authors discussed the alterations of lipid profiling of lipoproteins only in CVD and its outcomes. In that paper no methodological issues were discussed.

Kontush A, Chapman MJ. Lipidomics as a tool for the study of lipoprotein metabolism. Curr Atheroscler Rep. 2010 May;12(3):194-201.

Ding M, Rexrode KM. A Review of Lipidomics of Cardiovascular Disease Highlights the Importance of Isolating Lipoproteins. Metabolites. 2020 Apr 23;10(4):163.

Is the paper is well written? The format of the paper was well-organized and easy to follow, but the writing itself could be improved in terms of clarity and concision in some sections such as 4.2 for example the differences between pre-diabetes and diabetes is not clear or the role of Lipidomics in development of insulin resistance needs to be more explained.

According to the reviewer suggestion, changes were made in the section 4.2. in the revised manuscript.

Is the text clear and easy to read? Yes, in my opinion the text is easy to read.

I thank the reviewer for the nice comment.

Are the conclusions consistent with the evidence and arguments presented? Yes, in general. However, I would like to see the challenges that are exist for the Lipidomics research studies at the present.

Very interesting point. At present, many applications of lipidomic profiling remain confined to the research laboratory. Prospective clinical lipidomics platforms must demonstrate their clinical utility; that is, their ability to inform and impact patient diagnosis and care. Thus, a main challenge is to translate lipidomics into clinical application and to understand the importance of clinical lipidomics as one of the most helpful approaches during the design and decision making of therapeutic strategies for individuals. The clinical lipidomics should be merged with clinical phenomes, e.g., patient symptoms, biomedical analyses, pathology, images, and responses to therapies, although it is difficult to integrate and fuse the information of clinical lipidomics with clinical phenomes.

Do they address the main question posed? The main question that the author tried to address in the beginning of the paper was the shifts from traditional serum lipid measurements that are poorly descriptive of the total serum lipidomic profile to comprehensive lipid profiling is an urgent need for early diagnosing the metabolic syndromes and prevent cardio-metabolic disease. In my opinion the methodology and conclusion section of the paper are aligned with the main research question and it provides a convincing answer to the question that was posed

I thank the reviewer for the nice comments.

Reviewer 2 Report

1.     Some spelling mistakes should be addressed: such as in line 31, does “transported though” mean through? Line 39, does “Α cluster of metabolic diseases are trigged by” mean triggered? Line 52, “it has unique potential” should be changed as it has a unique potential.

2.     In part 2- From Conventional Lipid Biomarkers to Comprehensive Lipid Profiling, authors should list more references to address these differences, not only clinical research but also add some basic scientific research changes.

3.     The quality of Fig.2 is too low, please change or upload a better one.

4.     In part 4, the Cardiometabolic Diseases also include chronic renal failure (https://doi.org/10.1093/eurpub/cky112), please add more details and/ or related research status in this part.

Author Response

Reviewer 2

I thank the reviewer for the constructive comments. Please see my answers in red.

  1. Some spelling mistakes should be addressed: such as in line 31, does “transported though” mean through? Line 39, does “Α cluster of metabolic diseases are trigged by” mean triggered? Line 52, “it has unique potential” should be changed as it has a unique potential.

I thank the reviewer. All changes were made in the revised manuscript.

  1. In part 2- From Conventional Lipid Biomarkers to Comprehensive Lipid Profiling, authors should list more references to address these differences, not only clinical research but also add some basic scientific research changes.

It is not clear to the author whether a detailed description of the term “comprehensive lipid profiling” vs that of “conventional lipid biomarkers” is the point of interest. If so, changes have been made in the revised manuscript.

  1. The quality of Fig.2 is too low, please change or upload a better one.

In the revised manuscript, the quality of the Fig. 2 was improved.

  1. In part 4, the Cardiometabolic Diseases also include chronic renal failure (https://doi.org/10.1093/eurpub/cky112), please add more details and/ or related research status in this part.

According to the reviewer’s suggestion, related research on chronic renal failure was added in the revised manuscript.

Round 2

Reviewer 2 Report

Agree to accept!